# Large-Scale Stochastic Sampling from the Probability Simplex

**Jack Baker**
STOR-i CDT, Mathematics and Statistics
Lancaster University
j.baker1@lancaster.ac.uk

**Paul Fearnhead**
Mathematics and Statistics
Lancaster University
p.fearnhead@lancaster.ac.uk

**Emily B. Fox**
Computer Science & Engineering and Statistics
University of Washington
ebfox@uw.edu

**Christopher Nemeth**
Mathematics and Statistics
Lancaster University
c.nemeth@lancaster.ac.uk

## Abstract

Stochastic gradient Markov chain Monte Carlo (SGMCMC) has become a popular
method for scalable Bayesian inference. These methods are based on sampling a
discrete-time approximation to a continuous time process, such as the Langevin
diffusion. When applied to distributions defined on a constrained space the time-
discretization error can dominate when we are near the boundary of the space.
We demonstrate that because of this, current SGMCMC methods for the simplex
struggle with *sparse simplex spaces*; when many of the components are close to
zero. Unfortunately, many popular large-scale Bayesian models, such as network
or topic models, require inference on sparse simplex spaces. To avoid the biases
caused by this discretization error, we propose the stochastic Cox-Ingersoll-Ross
process (SCIR), which removes all discretization error and we prove that samples
from the SCIR process are asymptotically unbiased. We discuss how this idea can
be extended to target other constrained spaces. Use of the SCIR process within a
SGMCMC algorithm is shown to give substantially better performance for a topic
model and a Dirichlet process mixture model than existing SGMCMC approaches.

Stochastic gradient Markov chain Monte Carlo (SGMCMC) has become a popular method for
scalable Bayesian inference (Welling and Teh, 2011; Chen et al., 2014; Ding et al., 2014; Ma et al.,
2015). The foundation of SGMCMC methods are a class of continuous processes that explore a target
distribution—e.g., the posterior—using gradient information. These processes converge to a Markov
chain which samples from the posterior distribution exactly. SGMCMC methods replace the costly
full-data gradients with minibatch-based stochastic gradients, which provides one source of error.
Another source of error arises from the fact that the continuous processes are almost never tractable to
simulate; instead, discretizations are relied upon. In the non-SG scenario, the discretization errors are
corrected for using Metropolis-Hastings. However, this is not (generically) feasible in the SG setting.
The result of these two sources of error is that SGMCMC targets an approximate posterior (Welling
and Teh, 2011; Teh et al., 2016; Vollmer et al., 2016).

Another significant limitation of SGMCMC methods is that they struggle to sample from constrained
spaces. Naively applying SGMCMC can lead to invalid, or inaccurate values being proposed. The
result is large errors near the boundary of the space (Patterson and Teh, 2013; Ma et al., 2015; Li
et al., 2016). A particularly important constrained space is the simplex space, which is used to model
discrete probability distributions. A parameter $\omega$ of dimension $d$ lies in the simplex if it satisfies
the following conditions: $\omega_j \geq 0$ for all $j = 1, \ldots, d$ and $\sum_{j=1}^{d} \omega_j = 1$. Many popular models

contain simplex parameters. For example, latent Dirichlet allocation (LDA) is defined by a set of topic-specific distributions on words and document-specific distributions on topics. Probabilistic network models often define a link probability between nodes. More generally, mixture and mixed membership models have simplex-constrained mixture weights; even the hidden Markov model can be cast in this framework with simplex-constrained transition distributions. As models become large-scale, these vectors $\omega$ often become *sparse*–i.e., many $\omega_j$ are close to zero—pushing them to the boundaries of the simplex. All the models mentioned have this tendency. For example in network data, nodes often have relatively few links compared to the size of the network, e.g., the number of friends the average social network user has will be small compared with the size of the whole social network. In these cases the problem of sampling from the simplex space becomes even *harder*; since many values will be very close to the boundary of the space.

Patterson and Teh (2013) develop an improved SGMCMC method for sampling from the probability simplex: stochastic gradient Riemannian Langevin dynamics (SGRLD). The improvements achieved are through an astute transformation of the simplex parameters, as well as developing a Riemannian (see Girolami and Calderhead, 2011) variant of SGMCMC. This method achieved state-of-the-art results on an LDA model. However, we show that despite the improvements over standard SGMCMC, the discretization error of SGRLD still causes problems on the simplex. In particular, it leads to asymptotic biases which dominate at the boundary of the space and causes significant inaccuracy.

To counteract this, we design an SGMCMC method based on the Cox-Ingersoll-Ross (CIR) process. The resulting process, which we refer to as the stochastic CIR process (SCIR), has *no discretization error*. This process can be used to simulate from gamma random variables directly, which can then be moved into the simplex space using a well known transformation. The CIR process has a lot of nice properties. One is that the transition equation is known exactly, which is what allows us to simulate from the process without discretization error. We are also able to characterize important theoretical properties of the SCIR algorithm, such as the non-asymptotic moment generating function, and thus its mean and variance. We discuss how these ideas can be used to simulate efficiently from other constrained spaces, such as $(0, \infty)$.

We demonstrate the impact of this SCIR method on a broad class of models. Included in these experiments is the development of a scalable sampler for Dirichlet processes, based on the slice sampler of Walker (2007); Papaspiliopoulos (2008); Kalli et al. (2011). To our knowledge the application of SGMCMC methods to Bayesian nonparametric models has not been explored. All proofs in this article are relegated to the Supplementary Material. All code for the experiments is available online[1], and full details of hyperparameter and tuning constant choices has been detailed in the Supplementary Material.

# 1  Stochastic Gradient MCMC on the Probability Simplex

## 1.1  Stochastic Gradient MCMC

Consider Bayesian inference for continuous parameters $\theta \in \mathbb{R}^d$ based on data $\mathbf{x} = \{x_i\}_{i=1}^N$. Denote the density of $x_i$ as $p(x_i|\theta)$ and assign a prior on $\theta$ with density $p(\theta)$. The posterior is then defined, up to a constant of proportionality, as $p(\theta|\mathbf{x}) \propto p(\theta) \prod_{i=1}^N p(x_i|\theta)$, and has distribution $\pi$. We define $f(\theta) := -\log p(\theta|\mathbf{x})$. Whilst MCMC can be used to sample from $\pi$, such algorithms require access to the full data set at each iteration. Stochastic gradient MCMC (SGMCMC) is an approximate MCMC algorithm that reduces this per-iteration computational and memory cost by using only a small subset of data points at each step.

The most common SGMCMC algorithm is stochastic gradient Langevin dynamics (SGLD), first introduced by Welling and Teh (2011). This sampler uses the Langevin diffusion, defined as the solution to the stochastic differential equation

$$d\theta_t = -\nabla f(\theta_t)dt + \sqrt{2}dW_t, \tag{1.1}$$

where $W_t$ is a $d$-dimensional Wiener process. Similar to MCMC, the Langevin diffusion defines a Markov chain whose stationary distribution is $\pi$.

Unfortunately, simulating from (1.1) is rarely possible, and the cost of calculating $\nabla f$ is $O(N)$ since it involves a sum over all data points. The idea of SGLD is to introduce two approximations to

circumvent these issues. First, the continuous dynamics are approximated by discretizing them, in a similar way to Euler's method for ODEs. This approximation is known as the *Euler-Maruyama* method. Next, in order to reduce the cost of calculating $\nabla f$, it is replaced with a cheap, unbiased estimate. This leads to the following update equation, with user chosen stepsize $h$

$$\theta_{m+1} = \theta_m - h\nabla \hat{f}(\theta) + \sqrt{2h}\eta_m, \qquad \eta_m \sim N(0, 1). \tag{1.2}$$

Here, $\nabla \hat{f}$ is an unbiased estimate of $\nabla f$ whose computational cost is $O(n)$ where $n \ll N$. Typically, we set $\nabla \hat{f}(\theta) := -\nabla \log p(\theta) - N/n \sum_{i \in S_m} \nabla \log p(x_i|\theta)$, where $S_m \subset \{1, \dots, N\}$ resampled at each iteration with $|S_m| = n$. Applying (1.2) repeatedly defines a Markov chain that approximately targets $\pi$ (Welling and Teh, 2011). There are a number of alternative SGMCMC algorithms to SGLD, based on approximations to other diffusions that also target the posterior distribution (Chen et al., 2014; Ding et al., 2014; Ma et al., 2015).

Recent work has investigated reducing the error introduced by approximating the gradient using minibatches (Dubey et al., 2016; Nagapetyan et al., 2017; Baker et al., 2017; Chatterji et al., 2018). While, by comparison, the discretization error is generally smaller, in this work we investigate an important situation where it degrades performance considerably.

## 1.2 SGMCMC on the Probability Simplex

We aim to make inference on the simplex parameter $\omega$ of dimension $d$, where $\omega_j \geq 0$ for all $j = 1, \dots, d$ and $\sum_{j=1}^d \omega_j = 1$. We assume we have categorical data $z_i$ of dimension $d$ for $i = 1, \dots, N$, so $z_{ij}$ will be 1 if data point $i$ belongs to category $j$ and $z_{ik}$ will be zero for all $k \neq j$. We assume a Dirichlet prior $\text{Dir}(\alpha)$ on $\omega$, with density $p(\omega) \propto \prod_{j=1}^d \omega_d^{\alpha_j}$, and that the data is drawn from $z_i \,|\, \omega \sim \text{Categorical}(\omega)$ leading to a $\text{Dir}(\alpha + \sum_{i=1}^N z_i)$ posterior. An important transformation we will use repeatedly throughout this article is as follows: if we have $d$ random gamma variables $X_j \sim \text{Gamma}(\alpha_j, 1)$. Then $(X_1, \dots, X_d)/\sum_j X_j$ will have $\text{Dir}(\alpha)$ distribution, where $\alpha = (\alpha_1, \dots, \alpha_d)$.

In this simple case the posterior of $\omega$ can be calculated exactly. However, in the applications we consider the $z_i$ are latent variables, and they are also simulated as part of a larger Gibbs sampler. Thus the $z_i$ will change at each iteration of the algorithm. We are interested in the situation where this is the case, and $N$ is large, so that standard MCMC runs prohibitively slowly. The idea of SGMCMC in this situation is to use subsamples of $z$ to propose appropriate moves to $\omega$.

Applying SGMCMC to models which contain simplex parameters is challenging due to their constraints. Naively applying SGMCMC can lead to invalid values being proposed. The first SGMCMC algorithm developed specifically for the probability simplex was the SGRLD algorithm of Patterson and Teh (2013). Patterson and Teh (2013) try a variety of transformations for $\omega$ which move the problem onto a space in $\mathbb{R}^d$, where standard SGMCMC can be applied. They also build upon standard SGLD by developing a Riemannian variant. Riemannian MCMC (Girolami and Calderhead, 2011) takes the geometry of the space into account, which assists with errors at the boundary of the space. The parameterization Patterson and Teh (2013) find numerically performs best is $\omega_j = |\theta_j|/\sum_{j=1}^d |\theta_j|$. They use a mirrored gamma prior for $\theta_j$, which has density $p(\theta_j) \propto |\theta_j|^{\alpha_j - 1} e^{-|\theta_j|}$. This means the prior for $\omega$ remains the required Dirichlet distribution. They calculate the density of $z_i$ given $\theta$ using a change of variables and use a (Riemannian) SGLD update to update $\theta$.

## 1.3 SGRLD on Sparse Simplex Spaces

Patterson and Teh (2013) suggested that the boundary of the space is where most problems occur using these kind of samplers; motivating their introduction of Riemannian ideas for SGLD. In many popular applications, such as LDA and modeling sparse networks, many of the components $\omega_j$ will be close to 0. We refer to such $\omega$ as being *sparse*. In other words, there are many $j$ for which $\sum_{i=1}^N z_{ij} = 0$. In order to demonstrate the problems with using SGRLD in this case, we provide a similar experiment to Patterson and Teh (2013). We use SGRLD to simulate from a sparse simplex parameter $\omega$ of dimension $d = 10$ with $N = 1000$. We set $\sum_{i=1}^N z_{i1} = 800$, $\sum_{i=1}^N z_{i2} = \sum_{i=1}^N z_{i3} = 100$, and $\sum_{i=1}^N z_{ij} = 0$, for $3 < j \leq 10$. The prior parameter $\alpha$ was set to 0.1 for all components. Leading to

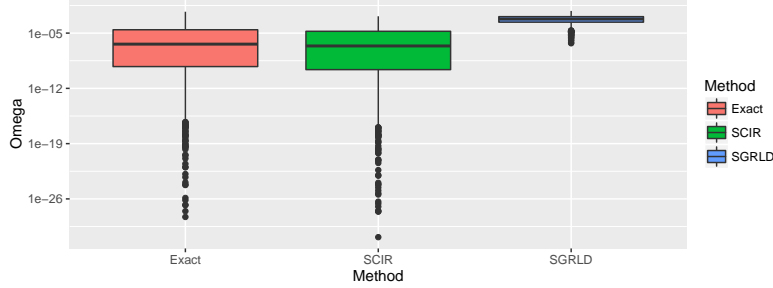

Figure 1: Boxplots of a 1000 iteration sample from SGRLD and SCIR fit to a sparse Dirichlet posterior, compared to 1000 exact independent samples. On the log scale.

a highly sparse Dirichlet posterior. We will refer back to this experiment as the *running experiment*. In Figure 1 we provide boxplots from a sample of the fifth component of $\omega$ using SGRLD after 1000 iterations with 1000 iterations of burn-in, compared with boxplots from an exact sample. The method SCIR will be introduced later. We can see from Figure 1 that SGRLD rarely proposes small values of $\omega$. This becomes a significant issue for sparse Dirichlet distributions, since the lack of small values leads to a poor approximation to the posterior, as we can see from the boxplots.

We hypothesize that the reason SGRLD struggles when $\omega_j$ is near the boundary is due to the discretization by $h$, and we now try to diagnose this issue in detail. The problem relates to the bias of SGLD caused by the discretization of the algorithm. We use the results of Vollmer et al. (2016) to characterize this bias for a fixed stepsize $h$. For similar results when the stepsize scheme is decreasing, we refer the reader to Teh et al. (2016). Proposition 1.1 is a simple application of Vollmer et al. (2016, Theorem 3.3), so we refer the reader to that article for full details of the assumptions. For simplicity of the statement, we assume that $\theta$ is 1-dimensional, but the results are easily adapted to the $d$-dimensional case.

**Proposition 1.1.** *(Vollmer et al., 2016) Under Vollmer et al. (2016, Assumptions 3.1 and 3.2), assume $\theta$ is 1-dimensional. Let $\theta_m$ be iteration $m$ of an SGLD algorithm for $m = 1, \ldots, M$, then the asymptotic bias defined by $\lim_{M \to \infty} \left| 1/M \sum_{m=1}^{M} \mathbb{E}[\theta_m] - \mathbb{E}_\pi[\theta] \right|$ has leading term $O(h)$.*

While ordinarily this asymptotic bias is hard to disentangle from other sources of error, as $\mathbb{E}_\pi[\theta]$ gets close to zero $h$ has to be set prohibitively small to give a good approximation to $\theta$. The crux of the issue is that, while the *absolute error* remains the same, at the boundary of the space the *relative error* is large since $\theta$ is small, and biased upwards due to the positivity constraint. To counteract this, in the next section we introduce a method which has no discretization error. This allows us to prove that the asymptotic bias, as defined in Proposition 1.1, will be zero for any choice of stepsize $h$.

## 2   The Stochastic Cox-Ingersoll-Ross Algorithm

We now wish to counteract the problems with SGRLD on sparse simplex spaces. First, we make the following observation: rather than applying a reparameterization of the prior for $\omega$, we can model the posterior for each $\theta_j$ directly and independently as $\theta_j \,|\, \mathbf{z} \sim \text{Gamma}(\alpha_j + \sum_{i=1}^{N} z_{ij}, 1)$. Then using the gamma reparameterization $\omega = \theta / \sum_j \theta_j$ still leads to the desired Dirichlet posterior. This leaves the $\theta_j$ in a much simpler form, and this simpler form enables us to remove all discretization error. We do this by using an alternative underlying process to the Langevin diffusion, known as the Cox-Ingersoll-Ross (CIR) process, commonly used in mathematical finance. A CIR process $\theta_t$ with parameter $a$ and stationary distribution $\text{Gamma}(a, 1)$ has the following form

$$d\theta_t = (a - \theta_t)dt + \sqrt{2\theta_t}dW_t. \tag{2.1}$$

The standard CIR process has more parameters, but we found changing these made no difference to the properties of our proposed scalable sampler, so we omit them (for exact details see the Supplementary Material).

The CIR process has many nice properties. One that is particularly useful for us is that the *transition density* is known exactly. Define $\chi^2(\nu, \mu)$ to be the non-central chi-squared distribution with $\nu$

degrees of freedom and non-centrality parameter $\mu$. If at time $t$ we are at state $\vartheta_t$, then the probability distribution of $\theta_{t+h}$ is given by

$$\theta_{t+h} \mid \theta_t = \vartheta_t \sim \frac{1 - e^{-h}}{2} W, \qquad W \sim \chi^2 \left( 2a, 2\vartheta_t \frac{e^{-h}}{1 - e^{-h}} \right). \qquad (2.2)$$

This transition density allows us to simulate directly from the CIR process with no discretization error. Furthermore, it has been proven that the CIR process is negative with probability zero (Cox et al., 1985), meaning we will not need to take absolute values as is required for the SGRLD algorithm.

## 2.1 Adapting for Large Datasets

The next issue we need to address is how to sample from this process when the dataset is large. Suppose that $z_i$ is data for $i = 1, \ldots, N$, for some large $N$, and that our target distribution is Gamma$(a, 1)$, where $a = \alpha + \sum_{i=1}^{N} z_i$. We want to approximate the target by simulating from the CIR process using only a subset of $\mathbf{z}$ at each iteration. A natural thing to do would be at each iteration to replace $a$ in the transition density equation (2.2) with an unbiased estimate $\hat{a} = \alpha + N/n \sum_{i \in S} z_i$, where $S \subset \{1, \ldots, N\}$, similar to SGLD. We will refer to a CIR process using unbiased estimates in this way as the *stochastic CIR process* (SCIR). Fix some stepsize $h$, which now determines how often $\hat{a}$ is resampled rather than the granularity of the discretization. Suppose $\hat{\theta}_m$ follows the SCIR process, then it will have the following update

$$\hat{\theta}_{m+1} \mid \hat{\theta}_m = \vartheta_m \sim \frac{1 - e^{-h}}{2} W, \qquad W \sim \chi^2 \left( 2\hat{a}_m, 2\vartheta_m \frac{e^{-h}}{1 - e^{-h}} \right), \qquad (2.3)$$

where $\hat{a}_m = \alpha + N/n \sum_{i \in S_m} z_i$.

We can show that this algorithm will approximately target the true posterior distribution in the same sense as SGLD. To do this, we draw a connection between the SCIR process and an SGLD algorithm, which allows us to use the arguments of SGLD to show that the SCIR process will target the desired distribution. More formally, we have the following relationship:

**Theorem 2.1.** *Let $\theta_t$ be a CIR process with transition 2.2. Then $U_t := g(\theta_t) = 2\sqrt{\theta_t}$ follows the Langevin diffusion for a generalized gamma distribution.*

Theorem 2.1, allows us to show that applying the transformation $g(\cdot)$ to the approximate SCIR process, leads to a discretization free SGLD algorithm for a generalized gamma distribution. Similarly, applying $g^{-1}(\cdot)$ to the approximate target of this SGLD algorithm leads to the desired Gamma$(a, 1)$ distribution. Full details are given after the proof of Theorem 2.1. The result means that similar to SGLD, we can replace the CIR parameter $a$ with an unbiased estimate $\hat{a}$ created from a minibatch of data. Provided we re-estimate $a$ from one iteration to the next using different minibatches, the approximate target distribution will still be Gamma$(a, 1)$. As in SGLD, there will be added error based on the noise in the estimate $\hat{a}$. However, from the desirable properties of the CIR process we are able to quantify this error more easily than for the SGLD algorithm, and we do this in Section 3.

Algorithm 1 below summarizes how SCIR can be used to sample from the simplex parameter $\omega \mid \mathbf{z} \sim \text{Dir}(\alpha + \sum_{i=1}^{N} \mathbf{z}_i)$. This can be done in a similar way to SGRLD, with the same per-iteration computational cost, so the improvements we demonstrate later are essentially for free.

---

**Algorithm 1:** Stochastic Cox-Ingersoll-Ross (SCIR) for sampling from the probability simplex.

---

**Input:** Starting points $\theta_0$, stepsize $h$, minibatch size $n$.
**Result:** Approximate sample from $\omega \mid \mathbf{z}$.
**for** $m = 1$ **to** $M$ **do**
    Sample minibatch $S_m$ from $\{1, \ldots, N\}$
    **for** $j = 1$ **to** $d$ **do**
        Set $\hat{a}_j \leftarrow \alpha + N/n \sum_{i \in S_m} z_{ij}$.
        Sample $\hat{\theta}_{mj} \mid \hat{\theta}_{(m-1)j}$ using (2.3) with parameter $\hat{a}_j$ and stepsize $h$.
    **end**
    Set $\omega_m \leftarrow \theta_m / \sum_j \theta_{mj}$.
**end**

---

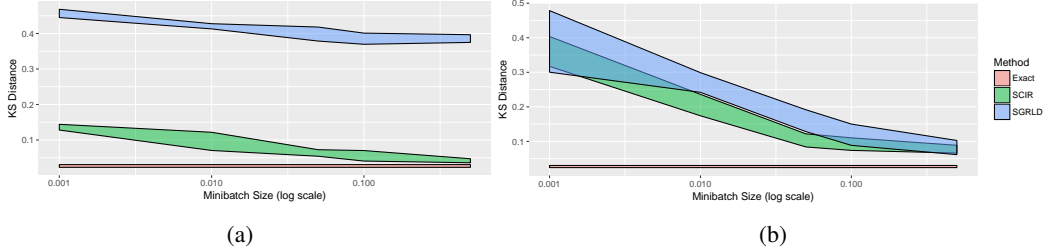

(a)&emsp;&emsp;&emsp;&emsp;&emsp;&emsp;&emsp;&emsp;&emsp;&emsp;&emsp;&emsp;&emsp;&emsp;(b)

Figure 2: Kolmogorov-Smirnov distance for SGRLD and SCIR at different minibatch sizes when used to sample from (a), a sparse Dirichlet posterior and (b) a dense Dirichlet posterior.

## 2.2 SCIR on Sparse Data

We test the SCIR process on two synthetic experiments. The first experiment is the running experiment on the sparse Dirichlet posterior of Section 1.3. The second experiment allocates 1000 datapoints equally to each component, leading to a highly dense Dirichlet posterior. For both experiments, we run 1000 iterations of optimally tuned SGRLD and SCIR algorithms and compare to an exact sampler. For the sparse experiment, Figure 1 shows boxplots of samples from the fifth component of $\omega$, which is sparse. For both experiments, Figure 2 plots the Kolmogorov-Smirnov distance ($d_{KS}$) between the approximate samples and the true posterior (full details of the distance measure are given in the Supplementary Material). For the boxplots, a minibatch of size 10 is used; for the $d_{KS}$ plots, the proportion of data in the minibatch is varied from 0.001 to 0.5. The $d_{KS}$ plots show the runs of five different seeds, which gives some idea of variability.

The boxplots of Figure 1 demonstrate that the SCIR process is able to handle smaller values of $\omega$ much more readily than SGRLD. The impact of this is demonstrated in Figure 2a, the sparse $d_{KS}$ plot. Here the SCIR process is achieving much better results than SGRLD, and converging towards the exact sampler at larger minibatch sizes. The dense $d_{KS}$ plot of Figure 2b shows that as we move to the dense setting the samplers have similar properties. The conclusion is that the SCIR algorithm is a good choice of simplex sampler for either the dense or sparse case.

## 2.3 Extensions

For simplicity, in this article we have focused on a popular usecase of SCIR: sampling from a $\text{Dir}(\alpha + \sum_{i=1}^{N} \mathbf{z}_i)$ distribution, with $\mathbf{z}$ categorical. This method can be easily generalized though. For a start, the SCIR algorithm is not limited to $\mathbf{z}$ being categorical, and it can be used to sample from most constructions that use Dirichlet distributions, provided the $\mathbf{z}$ are not integrated out. The method can also be used to sample from constrained spaces on $(0, \infty)$ that are gamma distributed by just sampling from the SCIR process itself (since the stationary distribution of the CIR process is gamma). There are other diffusion processes that have tractable transition densities. These can be exploited in a similar way to create other discretization free SGMCMC samplers. One such process is called geometric Brownian motion, which has lognormal stationary distribution. This process can be adapted to create a stochastic sampler from the lognormal distribution on $(0, \infty)$.

## 3 Theoretical Analysis

In the following theoretical analysis we wish to target a $\text{Gamma}(a, 1)$ distribution, where $a = \alpha + \sum_{i=1}^{N} z_i$ for some data $\mathbf{z}$. We run an SCIR algorithm with stepsize $h$ for $M$ iterations, yielding the sample $\hat{\theta}_m$ for $m = 1, \ldots, M$. We compare this to an exact CIR process with stationary distribution $\text{Gamma}(a, 1)$, defined by the transition equation in (2.2). We do this by deriving the moment generating function (MGF) of $\hat{\theta}_m$ in terms of the MGF of the exact CIR process. For simplicity, we fix a stepsize $h$ and, abusing notation slightly, set $\theta_m$ to be a CIR process that has been run for time $mh$.

**Theorem 3.1.** *Let $\hat{\theta}_M$ be the SCIR process defined in (2.3) starting from $\theta_0$ after $M$ steps with stepsize $h$. Let $\theta_M$ be the corresponding exact CIR process, also starting from $\theta_0$, run for time $Mh$,*

*and with coupled noise. Then the MGF of $\hat{\theta}_M$ is given by*

$$M_{\hat{\theta}_M}(s) = M_{\theta_M}(s) \prod_{m=1}^{M} \left[ \frac{1 - s(1 - e^{-mh})}{1 - s(1 - e^{-(m-1)h})} \right]^{-(\hat{a}_m - a)}, \tag{3.1}$$

*where we have*

$$M_{\theta_M}(s) = \left[ 1 - s(1 - e^{-Mh}) \right]^{-a} \exp\left[ \theta_0 \frac{s e^{-Mh}}{1 - s(1 - e^{-Mh})} \right].$$

The proof of this result follows by induction from the properties of the non-central chi-squared distribution. The result shows that the MGF of the SCIR can be written as the MGF of the exact underlying CIR process, as well as an error term in the form of a product. Deriving the MGF enables us to find the non-asymptotic bias and variance of the SCIR process, which is more interpretable than the MGF itself. The results are stated formally in the following Corollary.

**Corollary 3.2.** *Given the setup of Theorem 3.1,*

$$\mathbb{E}[\hat{\theta}_M] = \mathbb{E}[\theta_M] = \theta_0 e^{-Mh} + a(1 - e^{-Mh}).$$

*Since $\mathbb{E}_\pi[\theta] = a$, then $\lim_{M \to \infty} |\frac{1}{M} \sum_{m=1}^{M} \mathbb{E}[\hat{\theta}_m] - \mathbb{E}_\pi[\theta]| = 0$ and SCIR is asymptotically unbiased. Similarly,*

$$\mathbb{V}\mathrm{ar}[\hat{\theta}_M] = \mathbb{V}\mathrm{ar}[\theta_M] + (1 - e^{-2Mh}) \frac{1 - e^{-h}}{1 + e^{-h}} \mathbb{V}\mathrm{ar}[\hat{a}],$$

*where $\mathbb{V}\mathrm{ar}[\hat{a}] = \mathbb{V}\mathrm{ar}[\hat{a}_m]$ for $m = 1, \ldots, M$ and*

$$\mathbb{V}\mathrm{ar}[\theta_M] = 2\theta_0(e^{-Mh} - e^{-2Mh}) + a(1 - e^{-Mh})^2.$$

The results show that the approximate process is asymptotically unbiased. We believe this explains the improvements the method has over SGRLD in the experiments of Sections 2.2 and 4. We also obtain the non-asymptotic variance as a simple sum of the variance of the exact underlying CIR process, and a quantity involving the variance of the estimate $\hat{a}$. This is of a similar form to the strong error of SGLD (Sato and Nakagawa, 2014), though without the contribution from the discretization. The variance of the SCIR is somewhat inflated over the variance of the CIR process. Reducing this variance would improve the properties of the SCIR process and would be an interesting avenue for further work. Control variate ideas could be applied for this purpose (Nagapetyan et al., 2017; Baker et al., 2017; Chatterji et al., 2018) and they may prove especially effective since the mode of a gamma distribution is known exactly.

## 4  Experiments

In this section we empirically compare SCIR to SGRLD on two challenging models: latent Dirichlet allocation (LDA) and a Bayesian nonparametric mixture. Performance is evaluated by measuring the predictive performance of the trained model on a held out test set over five different seeds. Stepsizes and hyperparameters are tuned using a grid search over the predictive performance of the method. The minibatch size is kept fixed for both the experiments. In the Supplementary Material, we provide a comparison of the methods to a Gibbs sampler. This sampler is non-scalable, but will converge to the true posterior rather than an approximation. The aim of the comparison to Gibbs is to give the reader an idea of how the stochastic gradient methods compare to exact methods for the different models considered.

### 4.1  Latent Dirichlet Allocation

Latent Dirichlet allocation (LDA, see Blei et al., 2003) is a popular model used to summarize a collection of documents by clustering them based on underlying topics. The data for the model is a matrix of word frequencies, with a row for each document. LDA is based on a generative procedure. For each document $l$, a discrete distribution over the $K$ potential topics, $\theta_l$, is drawn as $\theta_l \sim \mathrm{Dir}(\alpha)$ for some suitably chosen hyperparameter $\alpha$. Each topic $k$ is associated with a discrete distribution $\phi_k$ over all the words in a corpus, meant to represent the common words associated with particular topics. This is drawn as $\phi_k \sim \mathrm{Dir}(\beta)$, for some suitable $\beta$. Finally, each word in document $l$ is drawn

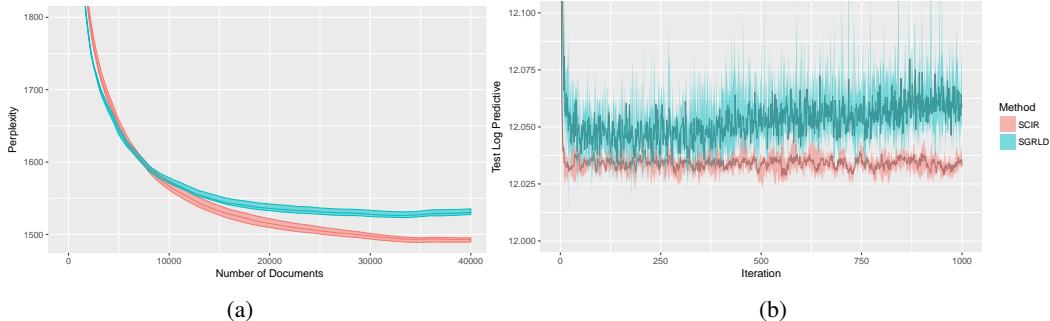

|  |  |
|---|---|
| (a) | (b) |

Figure 3: (a) plots the perplexity of SGRLD and SCIR when used to sample from the LDA model of Section 4.1 applied to Wikipedia documents; (b) plots the log predictive on a test set of the anonymous Microsoft user dataset, sampling the mixture model defined in Section 4.2 using SCIR and SGRLD.

a topic $k$ from $\theta_l$ and then the word itself is drawn from $\phi_k$. LDA is a good example for this method because $\phi_k$ is likely to be very sparse, there are many words which will not be associated with a given topic at all.

We apply SCIR and SGRLD to LDA on a dataset of scraped Wikipedia documents, by adapting the code released by Patterson and Teh (2013). At each iteration a minibatch of 50 documents is sampled in an online manner. We use the same vocabulary set as in Patterson and Teh (2013), which consists of approximately 8000 words. The exponential of the average log-predictive on a held out set of 1000 documents is calculated every 5 iterations to evaluate the model. This quantity is known as the perplexity, and we use a document completion approach to calculate it (Wallach et al., 2009). The perplexity is plotted for five runs using different seeds, which gives an idea of variability. Similar to Patterson and Teh (2013), for both methods we use a decreasing stepsize scheme of the form $h_m = h[1 + m/\tau]^{-\kappa}$. The results are plotted in Figure 3a. While the initial convergence rate is similar, SCIR keeps descending past where SGRLD begins to converge. This experiment illustrates the impact of removing the discretization error. We would expect to see further improvements of SCIR over SGRLD if a larger vocabulary size were used; as this would lead to sparser topic vectors. In real-world applications of LDA, it is quite common to use vocabulary sizes above 8000. The comparison to a collapsed Gibbs sampler, provided in the Supplementary Material, shows the methods are quite competetive to exact, non-scalable methods.

## 4.2 Bayesian Nonparametric Mixture Model

We apply SCIR to sample from a Bayesian nonparametric mixture model of categorical data, proposed by Dunson and Xing (2009). To the best of our knowledge, the development of SGMCMC methods for Bayesian nonparametric models has not been considered before. In particular, we develop a truncation free, scalable sampler based on SGMCMC for Dirichlet processes (DP, see Ferguson, 1973). For more thorough details of DPs and the stochastic sampler developed, the reader is referred to the Supplementary Material. The model can be expressed as follows

$$\mathbf{x}_i \,|\, \theta, z_i \sim \text{Multi}(n_i, \theta_{z_i}), \qquad \theta, z_i \sim \text{DP}(\text{Dir}(a), \alpha). \tag{4.1}$$

Here $\text{Multi}(m, \phi)$ is a multinomial distribution with $m$ trials and associated discrete probability distribution $\phi$; $\text{DP}(G_0, \alpha)$ is a DP with base distribution $G_0$ and concentration parameter $\alpha$. The DP component parameters and allocations are denoted by $\theta$ and $z_i$ respectively. We define the number of observations $N$ by $N := \sum_i n_i$, and let $L$ be the number of instances of $\mathbf{x}_i$, $i = 1, \ldots, L$. This type of mixture model is commonly used to model the dependence structure of categorical data, such as for genetic or natural language data (Dunson and Xing, 2009). The use of DPs means we can account for the fact that we do not know the true dependence structure. DPs allow us to learn the number of mixture components in a penalized way during the inference procedure itself.

We apply this model to the anonymous Microsoft user dataset (Breese et al., 1998). This dataset consists of approximately $N = 10^5$ instances of $L = 30000$ anonymized users. Each instance details part of the website the user visits, which is one of $d = 294$ categories (here $d$ denotes the dimension of $\mathbf{x}_i$). We use the model to try and characterize the typical usage patterns of the website. Since

there are a lot of categories and only an average of three observations for any one user, these data are expected to be sparse.

To infer the model, we devise a novel minibatched version of the slice sampler (Walker, 2007; Papaspiliopoulos, 2008; Kalli et al., 2011). We assign an uninformative gamma prior on $\alpha$, and this is inferred similarly to Escobar and West (1995). We minibatch the users at each iteration using $n = 1000$. For multimodal mixture models such as this, SGMCMC methods are known to get stuck in local modes (Baker et al., 2017), so we use a fixed stepsize for both SGRLD and SCIR. Once again, we plot runs over 5 seeds to give an idea of variability. The results are plotted in Figure 3b. They show that SCIR consistently converges to a lower log predictive test score, and appears to have lower variance than SGRLD. SGRLD also appears to be producing worse scores as the number of iterations increases. We found that SGRLD had a tendency to propose many more clusters than were required. This is probably due to the asymptotic bias of Proposition 1.1, since this would lead to an inferred model that has a higher $\alpha$ parameter than is set, meaning more clusters would be proposed than are needed. In fact, setting a higher $\alpha$ parameter appeared to alleviate this problem, but led to a worse fit, which is more evidence that this is the case.

In the Supplementary Material we provide plots comparing the stochastic gradient methods to the exact, but non-scalable Gibbs slice sampler (Walker, 2007; Papaspiliopoulos, 2008; Kalli et al., 2011). The comparison shows, while SCIR outperforms SGRLD, the scalable stochastic gradient approximation itself does not perform well in this case compared to the exact Gibbs sampler. This is to be expected for such a complicated model; the reason appears to be that the stochastic gradient methods get stuck in local stationary points. Improving the performance of stochastic gradient based samplers for Bayesian nonparametric problems is an important direction for future work.

## 5   Discussion

We presented an SGMCMC method, the SCIR algorithm, for simplex spaces. We show that the method has no discretization error and is asymptotically unbiased. Our experiments demonstrate that these properties give the sampler improved performance over other SGMCMC methods for sampling from sparse simplex spaces. Many important large-scale models are sparse, so this is an important contribution. A number of useful theoretical properties for the sampler were derived, including the non-asymptotic variance and moment generating function. We discuss how this sampler can be extended to target other constrained spaces discretization free. Finally, we demonstrate the impact of the sampler on a variety of interesting problems. An interesting line of further work would be reducing the non-asymptotic variance, which could be done by means of control variates.

## 6   Acknowledgments

Jack Baker gratefully acknowledges the support of the EPSRC funded EP/L015692/1 STOR-i Centre for Doctoral Training. Paul Fearnhead was supported by EPSRC grants EP/K014463/1 and EP/R018561/1. Christopher Nemeth acknowledges the support of EPSRC grants EP/S00159X/1 and EP/R01860X/1. Emily Fox acknowledges the support of ONR Grant N00014-15-1-2380 and NSF CAREER Award IIS-1350133.

## Footnotes

[1] Code available at `https://github.com/jbaker92/scir`.

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
