[Supplementary Material]

# A Proofs

## A.1 Proof of Proposition 1.1

*Proof.* Define the *local weak error* of SGLD, starting from $\theta_0$ and with stepsize $h$, with test function $\phi$ by

$$\mathbb{E}\left|\phi(\theta_1) - \phi(\bar{\theta}_h)\right|,$$

where $\bar{\theta}_h$ is the true underlying Langevin diffusion (1.1), run for time $h$ with starting point $\theta_0$. Then it is shown by Vollmer et al. (2016) that if $\phi : \mathbb{R}^d \to \mathbb{R}$ is a smooth test function, and that SGLD applied with test function $\phi$ has local weak error $O(h)$, then

$$\mathbb{E}\left|\lim_{M\to\infty} 1/M \sum_{m=1}^{M} \phi(\theta_m) - \mathbb{E}_\pi[\phi(\theta)]\right|$$

is also $O(h)$. What remains to be checked is that using such a simple function for $\phi$ (the identity), does not cause things to disappear such that the local weak error of SGLD is no longer $O(h)$. The identity function is infinitely differentiable, thus is sufficiently smooth. For SGLD, we find that

$$\mathbb{E}[\theta_1|\theta_0] = \theta_0 + hf'(\theta_0).$$

For the Langevin diffusion, we define the one step expectation using the weak Taylor expansion of Zygalakis (2011), which is valid since we have made Assumptions 3.1 and 3.2 of Vollmer et al. (2016). Define the infinitesimal operator $\mathcal{L}$ of the Langevin diffusion (1.1) by

$$\mathcal{L}\phi = f'(\theta) \cdot \partial_\theta\phi(\theta) + \partial_\theta^2\phi(\theta).$$

Then Zygalakis (2011) shows that the weak Taylor expansion of Langevin diffusion (1.1) has the form

$$\mathbb{E}[\bar{\theta}_h|\theta_0] = \theta_0 + h\mathcal{L}\phi(\theta_0) + \frac{h^2}{2}\mathcal{L}^2\phi(\theta_0) + O(h^3).$$

This means when $\phi$ is the identity then

$$\mathbb{E}[\bar{\theta}_h|\theta_0] = \theta_0 + hf'(\theta_0) + \frac{h^2}{2}\left[f(\theta)f'(\theta) + f''(\theta)\right] + O(h^3).$$

Since the terms agree up to $O(h)$ then it follows that even when $\phi$ is the identity, SGLD still has local weak error of $O(h)$. This completes the proof. $\qquad\square$

## A.2 Proof of Theorem 2.1

*Proof.* Suppose we have a random variable $U_\infty$ following a generalized gamma posterior with data $\mathbf{z}$ and the following density

$$f(u) \propto u^{2(\alpha + \sum_{i=1}^N z_i) - 1} e^{-u^2/4}.$$

Set $a := 2(\alpha + \sum_{i=1}^N z_i)$, Then $\partial \log f(u) = (2a-1)/u - u/2$, so that the Langevin diffusion for $U_\infty$ will have the following integral form

$$U_{t+h} \mid U_t = U_t + \int_t^{t+h} \left[\frac{2a-1}{U_s} - \frac{U_s}{2}\right] ds + \sqrt{2} \int_t^{t+h} dW_t.$$

Applying Ito's lemma to $U_t$ to transform to $\theta_t = g^{-1}(U_t) = U_t^2/4$ (here $g(\cdot)$ has been stated in the proof), we find that

$$\theta_{t+h} \mid \theta_t = \theta_t + \int_t^{t+h} [a - \theta_s] ds + \int_t^{t+h} \sqrt{2\theta_t} dW_t.$$

This is exactly the integral form for the CIR process. This completes the proof. $\qquad\square$

Now we give more details of the connection between SGLD and SCIR. Let us define an SGLD algorithm that approximately targets $U_\infty$, but without the Euler discretization by

$$U_{(m+1)h} \mid U_{mh} = U_{mh} + \int_{mh}^{(m+1)h} \left[\frac{2\hat{a}_m - 1}{U_s} - \frac{U_s}{2}\right] ds + \sqrt{2} \int_{mh}^{(m+1)h} dW_t, \tag{A.1}$$

where $\hat{a}_m$ is an unbiased estimate of $a$; for example, the standard SGLD estimate $\hat{a}_m = \alpha + N/n \sum_{i \in S_m} z_i$; also $h$ is a tuning constant which determines how much time is simulated before resampling $\hat{a}_m$.

Again applying Ito's lemma to $U_{mh}$ to transform to $\theta_{mh} = g(U_{mh}) = U_{mh}^2/4$, we find that

$$\theta_{(m+1)h} = \theta_{mh} + \int_{mh}^{(m+1)h} [\hat{a}_m - \theta_s]\, ds + \int_{mh}^{(m+1)h} \sqrt{2\theta_t}\, dW_t.$$

This is exactly the integral form for the update equation of an SCIR process.

Finally, to show SCIR has the desired approximate target, we use some properties of the gamma distribution. Firstly if $\theta_\infty \sim \text{Gamma}(a, 1)$ then $4\theta_\infty \sim \text{Gamma}(a, \frac{1}{4})$, so that $U_\infty = 2\sqrt{\theta_\infty}$ will have a generalized gamma distribution with density proportional to $h(u) \propto u^{2a-1}e^{-u^2/4}$. This is exactly the approximate target of the discretization free SGLD algorithm (A.1) we derived earlier.

### A.3  Proof of Theorem 3.1

First let us define the following quantities

$$r(s) = \frac{se^{-h}}{1 - s(1 - e^{-h})}, \qquad r^{(n)}(s) = \underbrace{r \circ \cdots \circ r}_{n}(s).$$

Then we will make use of the following Lemmas:

**Lemma A.1.** *For all $n \in \mathbb{N}$ and $s \in \mathbb{R}$*

$$r^{(n)}(s) = \frac{se^{-nh}}{1 - s(1 - e^{-nh})}.$$

**Lemma A.2.** *For all $n \in \mathbb{N}$, $s \in \mathbb{R}$, set $r^{(0)}(s) := s$, then*

$$\prod_{i=0}^{n-1} \left[1 - r^{(i)}(s)(1 - e^{-h})\right] = \left[1 - s(1 - e^{-nh})\right].$$

Both can be proved by induction, which is shown in Section B.

Suppose that $\theta_1|\theta_0$ is a CIR process, starting at $\theta_0$ and run for time $h$. Then we can immediately write down the MGF of $\theta_1$, $M_{\theta_1}(s)$, using the MGF of a non-central chi-squared distribution

$$M_{\theta_1}(s) = \mathbb{E}\left[e^{s\theta_1}|\theta_0\right] = \left[1 - s(1 - e^{-h})\right]^{-a} \exp\left[\frac{s\theta_0 e^{-h}}{1 - s(1 - e^{-h})}\right].$$

We can use this to find $\mathbb{E}\left[e^{s\theta_M}|\theta_{M-1}\right]$, and then take expectations of this with respect to $\theta_{M-2}$, i.e. $\mathbb{E}\left[\mathbb{E}\left[e^{s\theta_M}|\theta_{M-1}\right]|\theta_{M-2}\right]$. This is possible because $\mathbb{E}\left[e^{s\theta_M}|\theta_{M-1}\right]$ has the form $C(s)\exp[\theta_{M-1}r(s)]$, where $C(s)$ is a function only involving $s$, and $r(s)$ is as defined earlier. Thus repeatedly applying this and using Lemmas A.1 and A.2 we find

$$M_{\theta_M}(s) = \left[1 - s(1 - e^{-Mh})\right]^{-a} \exp\left[\frac{s\theta_0 e^{-Mh}}{1 - s(1 - e^{-Mh})}\right]. \tag{A.2}$$

Although this was already known, we can use the same idea to find the MGF of the SCIR process.

The MGF of SCIR immediately follows using the same logic as before, as well as using the form of $M_{\theta_M}(s)$ and Lemmas A.1 and A.2. Leading to

$$M_{\hat{\theta}_M}(s) = \prod_{m=1}^{M} \left[1 - r^{(m-1)}(s)(1 - e^{-h})\right]^{-\hat{a}_m} \exp\left[\theta_0 r^{(M)}(s)\right]$$

$$= M_{\theta_M}(s) \prod_{m=1}^{M} \left[\frac{1 - s(1 - e^{-mh})}{1 - s(1 - e^{-(m-1)h})}\right]^{-(\hat{a}_m - a)}.$$

## A.4 Proof of Theorem 3.2

*Proof.* From Theorem 3.1, we have

$$M_{\hat{\theta}_M}(s) = M_{\theta_M}(s) \underbrace{\prod_{m=1}^{M} \left[1 - s(1 - e^{-mh})\right]^{-(\hat{a}_m - a)}}_{e_0(s)} \underbrace{\prod_{m=1}^{M} \left[1 - s(1 - e^{-(m-1)h})\right]^{-(a - \hat{a}_m)}}_{e_1(s)}.$$

We clearly have $M_{\theta_M}(0) = e_0(0) = e_1(0) = 1$. Differentiating we find

$$e_0'(s) = \sum_{i=1}^{M} (\hat{a}_i - a)(1 - e^{-ih}) \left[1 - s(1 - e^{-ih})\right]^{-1} e_0(s),$$

similarly

$$e_1'(s) = \sum_{i=1}^{M} (a - \hat{a}_i)(1 - e^{-(i-1)h}) \left[1 - s(1 - e^{-(i-1)h})\right]^{-1} e_1(s).$$

It follows that, labeling the minibatch noise up to iteration $M$ by $\mathcal{B}_M$, and using the fact that $\mathbb{E}\hat{a}_i = a$ for all $i = 1, \dots, M$ we have

$$\begin{aligned}
\mathbb{E}\hat{\theta}_M &= \mathbb{E}\left[\mathbb{E}\left(\hat{\theta}_M | \mathcal{B}_M\right)\right] \\
&= \mathbb{E}\left[M_{\hat{\theta}_M}'(0)\right] \\
&= \mathbb{E}\left[M_{\theta_M}'(0)e_0(0)e_1(0) + M_{\theta_M}(0)e_0'(0)e_1(0) + M_{\theta_M}(0)e_0(0)e_1'(0)\right] \\
&= \mathbb{E}\theta_M.
\end{aligned}$$

Now taking second derivatives we find

$$e_0''(s) = \sum_{i=1}^{M} (\hat{a}_i - a)(\hat{a}_i - a - 1)(1 - e^{-ih})^2 \left[1 - s(1 - e^{-ih})\right]^{-2} e_0(s)$$

$$+ \sum_{i \neq j} (\hat{a}_i - a)(\hat{a}_j - a)(1 - e^{-ih})(1 - e^{-jh}) \left[1 - s(1 - e^{-ih})\right]^{-1} \left[1 - s(1 - e^{-jh})\right]^{-1} e_0(s).$$

Now taking expectations with respect to the minibatch noise, noting independence of $\hat{a}_i$ and $\hat{a}_j$ for $i \neq j$,

$$\mathbb{E}\left[e_0''(0)\right] = \sum_{i=1}^{M} (1 - e^{-ih})^2 \mathbb{V}\mathrm{ar}(\hat{a}_i).$$

By symmetry

$$\mathbb{E}\left[e_1''(0)\right] = \sum_{i=1}^{M} (1 - e^{-(i-1)h})^2 \mathbb{V}\mathrm{ar}(\hat{a}_i).$$

We also have

$$\mathbb{E}\left[e_0'(0)e_1'(0)\right] = -\sum_{i=1}^{M} (1 - e^{-ih})(1 - e^{-(i-1)h})\mathbb{V}\mathrm{ar}(\hat{a}_i).$$

Now we can calculate the second moment using the MGF as follows, note that $\mathbb{E}(e_0'(0)) = \mathbb{E}(e_1'(0)) = 0$,

$$
\begin{aligned}
\mathbb{E}\hat{\theta}_M^2 &= \mathbb{E}\left[M_{\hat{\theta}_M}''(0)\right] \\
&= \mathbb{E}\left[M_{\theta_M}''(0)e_0(0)e_1(0) + M_{\theta_M}(0)e_0''(0)e_1(0) + M_{\theta_M}(0)e_0(0)e_1''(0) + 2M_{\theta_M}(0)e_0'(0)e_1'(0)\right] \\
&= \mathbb{E}\theta_M^2 + \sum_{i=1}^{M}(1-e^{-ih})^2\mathbb{V}\text{ar}(\hat{a}_i) + \sum_{i=1}^{M}(1-e^{-(i-1)h})^2\mathbb{V}\text{ar}(\hat{a}_i) - 2\sum_{i=1}^{M}(1-e^{-ih})(1-e^{-(i-1)h})\mathbb{V}\text{ar}(\hat{a}_i) \\
&= \mathbb{E}\theta_M^2 + \mathbb{V}\text{ar}(\hat{a})\left[e^{-2Mh} - 1 + 2\sum_{i=1}^{M}\left(e^{-2(i-1)h} - e^{-(2i-1)h}\right)\right] \\
&= \mathbb{E}\theta_M^2 + \mathbb{V}\text{ar}(\hat{a})\left[e^{-2Mh} - 1 + 2\sum_{i=0}^{2M-1}(-1)^i e^{-ih}\right] \\
&= \mathbb{E}\theta_M^2 + \mathbb{V}\text{ar}(\hat{a})\left[e^{-2Mh} - 1 + \frac{2 - 2e^{-2Mh}}{1 + e^{-h}}\right] \\
&= \mathbb{E}\theta_M^2 + \mathbb{V}\text{ar}(\hat{a})(1 - e^{-2Mh})\left[\frac{1 - e^{-h}}{1 + e^{-h}}\right]
\end{aligned}
$$

$\square$

# B   Proofs of Lemmas

## B.1   Proof of Lemma A.1

*Proof.* We proceed by induction. Clearly the result holds for $n = 1$. Now assume the result holds for all $n \leq k$, we prove the result for $n = k + 1$ as follows

$$
\begin{aligned}
r^{(k+1)}(s) &= r \circ r^{(k)}(s) \\
&= r\left(\frac{se^{-kh}}{1 - s(1 - e^{-kh})}\right) \\
&= \frac{se^{-kh}}{1 - s(1 - e^{-kh})} \cdot \frac{e^{-h}(1 - s(1 - e^{-kh}))}{1 - s(1 - e^{-kh}) - se^{-kh}(1 - e^{-h})} \\
&= \frac{se^{-(k+1)h}}{1 - s(1 - e^{-(k+1)h})}.
\end{aligned}
$$

Thus the result holds for all $n \in \mathbb{N}$ by induction. $\square$

## B.2   Proof of Lemma A.2

*Proof.* Once again we proceed by induction. Clearly the result holds for $n = 1$. Now assume the result holds for all $n \leq k$. Using Lemma A.1, we prove the result for $n = k + 1$ as follows

$$
\begin{aligned}
\prod_{i=0}^{k}\left[1 - r^{(i)}(s)(1 - e^{-h})\right] &= \left[1 - s(1 - e^{-kh})\right]\left[1 - \frac{se^{-kh}(1 - e^{-h})}{1 - s(1 - e^{-kh})}\right] \\
&= \left[1 - s(1 - e^{-kh})\right]\left[\frac{1 - s(1 - e^{-(k+1)h})}{1 - s(1 - e^{-kh})}\right] \\
&= \left[1 - s(1 - e^{-(k+1)h})\right]
\end{aligned}
$$

Thus the result holds for all $n \in \mathbb{N}$ by induction. $\square$

## C  CIR Parameter Choice

As mentioned in Section 2, the standard CIR process has more parameters than those presented. The full form for the CIR process is as follows

$$d\theta_t = b(a - \theta_t)dt + \sigma\sqrt{\theta_t}dW_t, \tag{C.1}$$

where $a$, $b$ and $\sigma$ are parameters to be chosen. This leads to a $\text{Gamma}(2ab/\sigma^2, 2b/\sigma^2)$ stationary distribution. For our purposes, the second parameter of the gamma stationary distribution can be set arbitrarily, thus it is natural to set $2b = \sigma^2$ which leads to a $\text{Gamma}(a, 1)$ stationary distribution and a process of the following form

$$d\theta_t = b(a - \theta_t)dt + \sqrt{2b\theta_t}dW_t.$$

Fix the stepsize $h$, and use the slight abuse of notation that $\theta_m = \theta_{mh}$. The process has the following transition density

$$\theta_{m+1} \mid \theta_m = \vartheta_m \sim \frac{1 - e^{-bh}}{2}W, \qquad W \sim \chi^2\left(2a, 2\vartheta_m\frac{e^{-bh}}{1 - e^{-bh}}\right).$$

Using the MGF of a non-central chi-square distribution we find

$$M_{\theta_M}(s) = \left[1 - s(1 - e^{-Mbh})\right]^{-a}\exp\left[\frac{s\theta_0 e^{-Mbh}}{1 - s(1 - e^{-Mbh})}\right].$$

Clearly $b$ and $h$ are unidentifiable. Thus we arbitrarily set $b = 1$.

## D  Stochastic Slice Sampler for Dirichlet Processes

### D.1  Dirichlet Processes

The Dirichlet process (DP) (Ferguson, 1973) is parameterised by a scale parameter $\alpha \in \mathbb{R}_{>0}$ and a base distribution $G_0$ and is denoted $DP(G_0, \alpha)$. A formal definition is that $G$ is distributed according to $DP(G_0, \alpha)$ if for all $k \in \mathbb{N}$ and $k$-partitions $\{B_1, \ldots, B_k\}$ of the space of interest $\Omega$

$$(G(B_1), \ldots, G(B_k)) \sim \text{Dir}(\alpha G_0(B_1), \ldots, \alpha G_0(B_k)).$$

More intuitively, suppose we simulate $\theta_1, \ldots \theta_N$ from $G$. Then integrating out $G$ (Blackwell and MacQueen, 1973) we can represent $\theta_N$ conditional on $\theta_{-N}$ as

$$\theta_N \mid \theta_1, \ldots, \theta_{N-1} \sim \frac{1}{N - 1 + \alpha}\sum_{i=1}^{N-1}\delta_{\theta_i} + \frac{\alpha}{N - 1 + \alpha}G_0,$$

where $\delta_\theta$ is the distribution concentrated at $\theta$.

An explicit construction of a DP exists due to Sethuraman (1994), known as the *stick-breaking construction*. The slice sampler we develop in this section is based on this construction. For $j = 1, 2, \ldots$, set $V_j \sim \text{Beta}(1, \alpha)$ and $\theta_j \sim G_0$. Then the stick breaking construction is given by

$$\omega_j := V_j\prod_{k=1}^{j-1}(1 - V_k) \tag{D.1}$$

$$G \sim \sum_{j=1}^{\infty}\omega_j\delta_{\theta_j}, \tag{D.2}$$

and we have $G \sim DP(G_0, \alpha)$.

### D.2  Slice sampling Dirichlet process mixtures

We focus on sampling from Dirichlet process mixture models defined by

$$X_i \mid \theta_i \sim F(\theta_i)$$
$$\theta_i \mid G \sim G$$
$$G \mid G_0, \alpha \sim DP(G_0, \alpha).$$

A popular MCMC algorithm for sampling from this model is the *slice sampler*, originally developed by Walker (2007) and further developed by Papaspiliopoulos (2008); Kalli et al. (2011). The slice sampler is based directly on the stick-breaking construction (D.2), rather than the sequential (Pólya urn) formulation of (D.1). This makes it a more natural approach to develop a stochastic sampler from; since the stochastic sampler relies on conditional independence assumptions. The slice sampler can be extended to other Bayesian nonparametric models quite naturally, from their corresponding stick breaking construction.

We want to make inference on a Dirichlet process using the stick breaking construction directly. Suppose the mixture distribution $F$, and the base distribution $G_0$ admit densities $f$ and $g_0$. Introducing the variable $z$, which determines which component $x$ is currently allocated to, we can write the density as follows

$$p(x|\omega, \theta, z) \propto \omega_z f(x|\theta_z).$$

Theoretically we could now use a Gibbs sampler to sample conditionally from $z$, $\theta$ and $\omega$. However this requires updating an infinite number of weights, similarly $z$ is drawn from a categorical distribution with an infinite number of categories. To get around this Walker (2007) introduces another latent variable $u$, such that the density is now

$$p(x|\omega, \theta, z, u) \propto \mathbf{1}(u < \omega_z) f(x|\theta_z),$$

so that the full likelihood is given by

$$p(\mathbf{x}|\omega, \theta, \mathbf{z}, \mathbf{u}) \propto \prod_{i=1}^{N} \mathbf{1}(u_i < \omega_{z_i}) f(x_i|\theta_{z_i}). \tag{D.3}$$

Walker (2007) shows that in order for a standard Gibbs sampler to be valid given (D.3), the number of weights $\omega_j$ that needs to be sampled given this new latent variable is now finite, and given by $k^*$, where $k^*$ is the smallest value such that $\sum_{j=1}^{k^*} \omega_j > 1 - u_i$.

The Gibbs algorithm can now be stated as follows, note we have included an improvement suggested by Papaspiliopoulos (2008), in how to sample $v_j$.

- Sample the slice variables $\mathbf{u}$, given by $u_i \mid \omega, \mathbf{z} \sim U(0, \omega_{z_i})$ for $i = 1, \ldots, N$. Calculate $u^* = \min \mathbf{u}$.

- Delete or add components until the number of current components $k^*$ is the smallest value such that $u^* < 1 - \sum_{j=1}^{k^*} \omega_j$.

- Draw new component allocations $z_i$ for $i = 1, \ldots, N$, using $p(z_i = j|x_i, u_i, \omega, \theta) \propto \mathbf{1}(\omega_j > u_i) f(x_i|\theta)$.

- For $j \leq k^*$, sample new component parameters $\theta_j$ from $p(\theta_j|\mathbf{x}, \mathbf{z}) \propto g_0(\theta_j) \prod_{i \,:\, z_i = j} f(x_i|\theta_j)$

- For $j \leq k^*$ calculate simulate new stick breaks $v$ from $v_j \mid \mathbf{z}, \alpha \sim \text{Beta}\left(1 + m_j, \alpha + \sum_{l=j+1}^{k^*} m_l\right)$. Here $m_j := \sum_{i=1}^{N} \mathbf{1}_{z_i = j}$.

- Update $\omega$ using the new $v$: $\omega_j = v_j \prod_{l < j} (1 - v_j)$.

### D.3 Stochastic Sampler

The conditional independence of each update of the slice sampler introduced in Section D.2 makes it possible to adapt it to a stochastic variant. Suppose we update $\theta$ and $v$ given a minibatch of the $\mathbf{z}$ and $\mathbf{u}$ parameters. Then since the $\mathbf{z}$ and $\mathbf{u}$ parameters are just updated from the marginal of the posterior, only updating a minibatch of these parameters at a time would leave the posterior as the invariant distribution. Our exact MCMC procedure is similar to that in the R package PReMiuM (Liverani et al., 2015), though they do not use a stochastic sampler. First define the following: $Z^* = \max \mathbf{z}$; $S \subset \{1, \ldots, N\}$ is the current minibatch; $u^* = \min \mathbf{u}_S$; $k^*$ is the smallest value such that $\sum_{j=1}^{k^*} \omega_j > 1 - u^*$. Then our updates proceed as follows:

- Recalculate $Z^*$ and $S$ (note this can be done in $O(n)$ time since only $n$ $\mathbf{z}$ values changed).

Figure 4: (a) plots the perplexity of SGRLD, SCIR and Gibbs when used to sample from the LDA model of Section 4.1 applied to Wikipedia documents; (b) plots the log predictive on a test set of the anonymous Microsoft user dataset, sampling the mixture model defined in Section 4.2 using SCIR, SGRLD and Gibbs.

- For $j = 1, \ldots, Z^*$ sample $v_j$ stochastically with SCIR from
  $v_j \,|\, \mathbf{z}, \alpha \sim \mathrm{Beta}(1 + \hat{m}_j, \alpha + \sum_{l=j+1}^{k^*} \hat{m}_l)$. Here $\hat{m}_j = N/n \sum_{i \in S} \mathbf{1}_{z_i = j}$.
- Update $\omega_j$ using the new $v$: $\omega_j = v_j \prod_{l < j}(1 - v_j)$.
- For $j = 1, \ldots, Z^*$ sample $\theta_j$ stochastically with SGMCMC from
  $p(\theta_j | \mathbf{x}, \mathbf{z}) \propto g_0(\theta_j) \prod_{S_j} f(x_i | \theta_j)$. Here $S_j = \{i : z_i = j \text{ and } i \in S\}$.
- For $i \in S$ sample the slice variables $u_i \,|\, \omega, \mathbf{z} \sim U(0, \omega_{z_i})$.
- Sample $\alpha$ if required. Using Escobar and West (1995), for our example we assume a Gamma$(b_1, b_2)$ prior so that $\alpha \,|\, v_{1:Z^*} \sim \mathrm{Gamma}(b_1 + Z^*, b_2 - \sum_{j=1}^{K^*} \log(1 - v_j))$.
- Recalculate $u^*$. Sample additional $\omega_j$ from the prior, until $k^*$ is reached. For $j = (Z^* + 1), \ldots, k^*$ sample additional $\theta_j$ from the prior.
- For $i \in S$, sample $z_i$, where $\mathbb{P}(z_i = j | u_i, \omega, \theta, \mathbf{x}) \propto \mathbf{1}(\omega_j > u_i) f(x_i | \theta_j)$.

Note that for our particular example, we have the following conditional update for $\theta$ (ignoring minibatching for simplicity):

$$\theta_j \,|\, z_j, \mathbf{x} \sim \mathrm{Dirichlet}\left(a + \sum_{i \in S_j} x_{i1}, \ldots, a + \sum_{i \in S_j} x_{id}\right).$$

# E   Experiments

## E.1   Comparison with Gibbs

We provide a comparison of the SGRLD and SCIR algorithms for both experiments to an exact, but non-scalable Gibbs sampler. Figure 4a compares SGRLD and SCIR run on the LDA model to an exact collapsed Gibbs sampler (Griffiths and Steyvers, 2004), run for 100 iterations. Although due to the large-scale dataset, it was not possible to run the Gibbs algorithm for very many iterations, it shows that the SCIR algorithm for LDA is competetive to exact, non-scalable methods.

Figure 4b compares the SGRLD and SCIR algorithms to the Gibbs slice sampler of Walker (2007); Papaspiliopoulos and Roberts (2008); Kalli et al. (2011), run until convergence. While SCIR outperforms SGRLD, the methods are not that competetive with the Gibbs sampler. This is to be expected, since stochastic gradient methods converge only to an approximation of the posterior, while the Gibbs sampler converges to the true posterior. The reason the stochastic gradient methods do particularly badly in this case is due to the methods getting stuck in local stationary points. Fitting Bayesian nonparametric models at scale remains a challenging problem, and further work which improves the performance of these scalable samplers would be useful. The hyperparameters used for the Gibbs sampler is given in the tables in the sections below.

| Method | h | | | | | | |
|--------|-----|------|------|------|------|------|------|
| SCIR | 1.0 | 5e-1 | 1e-1 | 5e-2 | 1e-2 | 5e-3 | 1e-3 |
| SGRLD | 5e-1 | 1e-1 | 5e-2 | 1e-2 | 5e-3 | 1e-3 | 5e-4 | 1e-4 |

Table 1: Stepsizes for the synthetic experiment

| Method | $h$ | $\tau$ | $\kappa$ | $\alpha$ | $\beta$ | $K$ | $n$ | Gibbs Samples |
|--------|------|-------|------|-------|--------|-----|-----|---------------|
| CIR | 0.5 | 10. | .33 | 0.1 | 0.5 | 100 | 50 | 200 |
| SGRLD | 0.01 | 1000. | .6 | 0.01 | 0.0001 | 100 | 50 | 200 |
| Gibbs | | | | 0.1 | 0.5 | 100 | | |

Table 2: Hyperparameters for the LDA experiment

| Method | $h_\theta$ | $h_{\text{DP}}$ | $a$ | $K$ | $n$ |
|--------|-----------|----------------|-------|-----|------|
| CIR | 0.1 | 0.1 | 0.5 | 20 | 1000 |
| SGRLD | 0.001 | 0.005 | 0.001 | 30 | 1000 |
| Gibbs | | | 0.5 | | |

Table 3: Hyperparameters for the Bayesian nonparametric mixture experiment

## E.2 Synthetic

We now fully explain the distance measure used in the synthetic experiments. Suppose we have random variables $X$ taking values in $\mathbb{R}$ with cumulative density function (CDF) $F$. We also have an approximate sample from $X$, $\hat{X}$ with empirical density function $\hat{F}$. The Kolmogorov-Smirnov distance $d_{KS}$ between $X$ and $\hat{X}$ is defined by $d_{KS}(X, \hat{X}) = \sup_{x \in \mathbb{R}} \left\| \hat{F}(x) - F(x) \right\|$. However the Dirichlet distribution is multi-dimensional, so we measure the average Kolmogorov-Smirnov distance across dimensions by using the Rosenblatt transform (Rosenblatt, 1952).

Suppose now that $X$ takes values in $\mathbb{R}^d$. Define the conditional CDF of $X_k = x_k | X_{k-1} = x_{k-1}, \ldots, X_1 = x_1$ to be $F(x_k | \mathbf{x}_{1:(k-1)})$. Suppose we have an approximate sample from $X$, which we denote $\mathbf{x}^{(m)}$, for $m = 1, \ldots M$. Define $\hat{F}_j$ to be the empirical CDF defined by the samples $F(x_j^{(m)} | \mathbf{x}_{1:(j-1)}^{(m)})$. Then Rosenblatt (1952) showed that if $\hat{X}$ is a true sample from $X$ then $\hat{F}_j$ should be the uniform distribution and independent of $\hat{F}_k$ for $k \neq j$. This allows us to define a Kolmogorov-Smirnov distance measure across multiple dimensions as follows

$$d_{KS}(X, \hat{X}) = \frac{1}{K} \sum_{j=1}^{K} \sup_{x \in \mathbb{R}} \left\| \hat{F}_j(x) - F_j(x) \right\|.$$

Where here applying Rosenblatt (1952), $F_j(X)$ is just the uniform distribution.

The full posterior distributions for the sparse and dense experiments are as follows:

$$\omega_{\text{sparse}} \,|\, \mathbf{z} \sim \text{Dir} \left[ 800.1, 100.1, 100.1, 0.1, 0.1, 0.1, 0.1, 0.1, 0.1, 0.1 \right],$$
$$\omega_{\text{dense}} \,|\, \mathbf{z} \sim \text{Dir} \left[ 112.1, 119.1, 92.1, 98.1, 95.1, 96.1, 102.1, 92.1, 91.1, 103.1 \right].$$

For each of the five random seeds, we pick the stepsize giving the best $d_{KS}$ for SGRLD and SCIR from the options given in Table 1.

## E.3 Latent Dirichlet Allocation

As mentioned in the main body, we use a decreasing stepsize scheme of the form $h_m = h(1+m/\tau)^{-\kappa}$. We do this to be fair to SGRLD, where the best performance is found by using this decreasing scheme (Patterson and Teh, 2013; Ma et al., 2015); and this will probably reduce some of the bias due to the stepsize $h$. We find a decreasing stepsize scheme of this form also benefits SCIR, so we use it as well. Notice that we find similar optimal hyperparameters for SGRLD to Patterson and Teh (2013). Table 2 fully details the hyperparameter settings we use for the LDA experiment.

### E.4 Bayesian Nonparametric Mixture

For details of the stochastic slice sampler we use, please refer to Section D. Table 3 details full hyper-parameter settings for the Bayesian nonparametric mixture experiment. Note that $h_\theta$ corresponds to the stepsizes assigned for sampling the $\theta$ parameters; while $h_{DP}$ corresponds to the stepsizes assigned for sampling from the weights $\omega$ for the Dirichlet process.