[Reviews · NeurIPS 2018]

Reviewer 1



The authors first note that the existing SGLD techniques (derived from the SGLD) are not appropriate to sample Dirichlet distribution on the probability simplex. This is not a big surprise, SGLD is a general purpose sampler, which has been designed to sample from generic smooth distribution over $\mathbb{R}^d$. Some"generic" modifications can be brought to the SGLD to sample from a convex subset, but of course these methods are generic and are not well adapted to sample "sparse" Dirichlet distribution. I fully agree with these conclusions. In a second part, the authors proposed a new algorithm to sample from "sparse" Dirichlet distribution. It is well known that sampling a d-dimensional Dirichlet distribution is equivalent to sampling d independent Gamma random variables. There are many computationally efficient methods to sample independent Gamma distributions, and therefore sampling large dimensional Dirichlet distribution. The authors suggest to use independent Cox-Ingersol-Ross whose stationary distribution is known to be a gamma distribution. The rationale of using such transform stems from the fact that it can be adapted to the "big data" context (exactly along the same lines than the SGLD). Convergence results are presented in Section 4 and sketch of proofs (the main arguments are given, nevertheless the proofs are concise and not really to follow) are presented in the supplementary material. The paper is supported by two experiments, showing the clear superiority of the proposed approach wrt to SGLD. It would have been of interest to compare with the results of a state-of-the art sampler on this problem. Otherwise, this is clearly a solid piece of work, which would deserve to be accepted.

Reviewer 2



This paper introduces the CIR/SCIR sampler, which allows to draw samples from the simplex for Dirichlet/multinomial models. In contrast to previous samplers, CIR/SCIR can draw accurate samples in sparse scenarios. The papers shows the superiority of CIR/SCIR on toy examples, as well as on LDA and DPMM. The paper is very well written and the comparison to the literature seems fair. My first comment is the following. There is a stochastic process of the form (3.1) for each component theta. Why are all of these processes independent? I would have expected these processes to depend one another. Am I missing something? The method presented allows to obtain samples in the simplex for cases where there is a (multinomial or categorical) discrete variable next. Would the same approach be valid for a Dirichlet/Dirichlet construction, or for a more general class of models? If not, the assumptions about the model should be stated early in the paper. Related to the previous question, it seems like CIR/SCIR requires a block-sampling scheme in which the discrete variables z are explicitly instantiated. Isn't this a limitation compared to an alternative approach that integrates out z and performs HMC (or a similar MCMC method) on the remaining latent variables? Finallly, how are the parameters of SCIR tuned? For example, the parameter h in (3.3). In line 211, the paper states that the algorithms are "optimally tuned", but there aren't further details on this.

Reviewer 3



For the valuable problem of large-scale and sparse stochastic inference on simplex, the authors proposed a novel Stochastic gradient Markov chain Monte Carlo (SGMCMC) method, which is based on the Cox-Ingersoll-Ross (CIR) process. Compared with the commonly-used Langevin diffusion within the SGMCMC community, the CIR process (i) is closely related to the flexible Gamma distribution, and therefore more suitable for inferring a Dirichlet distribution on simplex, since a Dirichlet distribution is just the normalization of Gamma distributions; (ii) CIR has no discretization error, which is shown to be a clear advantage over the Langevin diffusion on simplex inference. Besides, the author proved that the proposed SCIR method is asymptotically unbiased, and has improved performance over other SGMCMC method on sparse simplex problem via two experiments, namely inferring a LDA on a dataset of scraped Wikipedia documents and inferring a Bayesian nonparametric mixture model on Microsoft user dataset. I think the quality is good; the presentation is clear; as far as I know the proposed technique is original and of great significance. Therefore I vote for acceptance. However, the experiments are okay, but not strong. More experiments would be better. I have read the authors' response, I would suggest additional experiments, such as exploiting the presented SCIR for more complicated non-conjugate models like the PGBN [1]; (2) to make a direct comparison with the TLASGR-MCMC developed in [2], which is also developed for inference on simplex and is shown better than the SGRLD. Other potential reference methods include the ones in [3,4]. [1] The Poisson gamma belief network. NIPS 2015. [2] Deep latent Dirichlet allocation with topic-layer-adaptive stochastic gradient Riemannian MCMC. ICML 2017. [3] "A Complete Recipe for Stochastic Gradient MCMC. NIPS 2015" [4] "Stochastic gradient geodesic MCMC methods. NIPS 2016"